# Estonian Parents’ Awareness of Pediculosis and Its Occurrence in Their Children

**DOI:** 10.3390/medicina58121773

**Published:** 2022-11-30

**Authors:** Ave Kutman, Ülle Parm, Anna-Liisa Tamm, Birgit Hüneva, Diana Jesin

**Affiliations:** Department of Physiotherapy and Environmental Health, Tartu Health Care College, 50411 Tartu, Estonia

**Keywords:** pediculosis, head lice, children, knowledge, Estonian

## Abstract

*Background and Objectives:* Pediculosis, or head lice infestation, is a widespread health problem that can affect anyone, regardless of gender, age, or social background. The purpose of this study was to clarify the occurrence of pediculosis among Estonian preschool- and primary school-aged children according to their parents and the parent’s awareness of pediculosis and related behaviors. *Materials and Methods:* An online questionnaire was completed by the parents of the preschool children (*n* = 1141) in 2019 and the parents of the elementary school children (*n* = 362) in 2021. For the descriptive data, *t*-test, Mann–Whitney or χ^2^ test, linear regression, and logistic regression analyses were applied. *Results:* According to the parents, pediculosis had occurred in 34.7% of the children, and more than one-third of pediculosis patients had experienced it more than twice. Lice were mainly acquired from elementary school or preschool and less often from friends, relatives, or training environments. Parents’ knowledge of head lice was rather good; the average score of the correct answers was 14.0 ± 3.4 (max. 20). In the multivariate analysis, higher age (coefficient 0.07, *p* < 0.001), healthcare education (coefficient 1.19, *p* < 0.001), and a previous occurrence of pediculosis in a family (coefficient 1.95; *p* < 0.001) were factors influencing better knowledge. In order to treat the infestation, antilice shampoo and combing were the most often used methods. *Conclusion:* Despite parents’ awareness, pediculosis infestations continue to be common among our children.

## 1. Introduction

Pediculosis is an infestation of head lice (*Pediculus humanus capitis*), including eggs and larvae. The condition usually causes severe itching and can lead to skin inflammation on the scalp [1]. Head lice are highly contagious and are transmitted from person to person, directly or indirectly, through contact with objects that are infested with lice, such as hats, brushes, combs, towels, and clothing [2]. Pediculosis affects millions of people worldwide each year and has been reported in all countries and in all socioeconomic classes [3]. Different studies have shown that the worldwide rates of head lice infestation vary between 0.7 and 59%, and in Europe, between 0.48 and 22.4% [4]. It has previously been shown that socioeconomic status, hair length, urban and rural location, family size, number of people in a home, and the sharing of items (combs, hats, hairpins, and clips) can all affect the transmission of head lice [5]. Children between the ages of six and nine are most often infected [6], with girls affected more than boys [7,8]. The correct treatment of pediculosis by parents is critical to the successful elimination of outbreaks, primarily because households are considered to be one of the most effective sites of infestation management [9]. However, misdiagnosis and the overuse of pediculicides increase resistance to treatment [10].

The transmission of head lice is also a widespread health problem in Estonia. According to the Estonian Health Board, in recent years (2016–2019), the prevalence of pediculosis has increased, primarily in children and welfare institutions, although statistics on infestations have not been kept since 2004. As a result, there is currently no accurate overview of how many preschool and school-age children have been infested, although the Health Board has received complaints from schools about the spread of head lice among children, which shows that the infestations have not disappeared and still need attention [11].

Most of the reports in the literature deal with the prevalence and treatment [9,12,13,14,15] of pediculosis. Sidoti et al. [10] argued that less had been done to assess knowledge about head lice in parents or families. A few studies on pediculosis across the world have shown that, in general, parents’ knowledge of transmission, treatment, and control of head lice infestations is limited. A survey of 1338 parents [16] found that only 7.1% of respondents correctly answered all 10 questions about lice recognition, prevention, and treatment, and more than one-third failed to correctly answer half of the questions. Another study about parents’ knowledge of pediculosis found that 32.9% of respondents (*n* = 237) knew the correct information about head lice transmission, but 40.1% of the respondents mistakenly correlated the control of pediculosis with hygiene. In a different survey, more than 90% of the participants responded correctly about head lice transmission [17].

In Nigeria, 496 people were examined for the presence of head lice and their awareness of its transmission. The results showed that 74% of the respondents had suffered from pediculosis, but only 11.1% of them knew how head lice are transmitted. Combing was most often mentioned as a treatment method (46.3%), and only 4.6% of the respondents had used lice medications [18]. Another survey about parents’ awareness showed that certain beliefs and perceptions regarding head lice generated worry and confusion in parents, as well as unhealthy actions to combat infestations. It was also revealed that there was an inverse correlation between parents’ educational level and the number of pediculosis cases [19]. It has also been found that maternal education, knowledge, and attitude toward pediculosis capitis infestation are significantly associated with the prevalence of pediculosis [8]. The same factors, and also the number of times combing hair per day, were pointed out as risk factors for pediculosis in another study [20]. One study revealed a greater association between the prevalence of infestations and individuals who already had a previous history of pediculosis and who had suffered from head itching. There was a significant relationship between pediculosis, head itching, and a previous history of pediculosis [21]. The previously mentioned studies were carried out in Africa, South America, or the Middle East, and thus one may doubt whether these results are suitable for the Estonian cultural space. For example, the results of studies conducted in Northern Europe (Norway) show that lice may be less common among children of the same age but are still abundant (about 45% of children) [22], and, in addition to combing, pediculicides are also used to treat pediculosis, but parents’ awareness of pediculosis is still important for identifying and dealing with lice [23]. Therefore, it seems that regardless of geographic location, the factors associated with lice are still similar as they are highly contagious, regardless of country.

In an attempt to better understand why head lice are still so common, particularly among children, our study examined parents’ knowledge of pediculosis in Estonia. This research aimed at detecting the level of pediculosis in preschool and elementary school children and parents’ knowledge, understanding, and possible related behaviors associated with the perception of pediculosis. The corresponding results are extremely necessary at the national level for the Board of Health, which can plan further activities according to parents’ awareness or lack of knowledge (for example, restoring the registration of pediculosis cases), as well as for family doctors and family nurses, whose task is primarily to counsel parents (including about pediculosis).

## 2. Materials and Methods

The study was carried out in two stages for the whole country of Estonia (nonprobability sampling). The targeted participants for the first stage were all parents in Estonia who had at least one preschooler (child’s expected age from 0 to 6 years); the targeted participants for the second stage were parents who had at least one child in elementary school (child’s expected age up to 10 years). Data from the first stage of the study were collected from December 2019 to February 2020; data for the second stage of the study were collected from October 2020 to February 2021. Ethical approval for both stages of the study was granted by the Research Ethics Committee of the University of Tartu (297/T-5, 29 October 2019; 332/M-18, 23 November 2020).

The questionnaire was prepared using Google Forms and made publicly accessible via social media. The questionnaire was distributed in internet environments, such as Facebook, as well as among family school forums. The questionnaire consisted of two parts: (1) the respondent’s general data (gender, age, education, and county), the presence of head lice in a child, and the treatment methods used (open question); (2) the respondent’s knowledge of the head lice. If a respondent’s child had not had head lice, the respondent moved on to the knowledge part of the questionnaire, skipping those questions about the transmission and treatment of lice.

The first part of the questionnaire was compiled by the surveyors in consultation with the then Deputy Director General at Estonian Health Board (Jelena Tomasova, 2019). Data referring to knowledge in relation to the phenomenon of pediculosis were obtained through 20 statements with multiple-choice answers (true, false, or unsure). The statements had been formulated and previously used by Professor ALBashtawy in his research [24], and permission was also requested to use them for conducting our study. These statements focused on some of the common myths associated with head lice transmission, treatment, and management. One statement was replaced due to cultural differences (the statement was about contacting a medical doctor before using a hair lice product, but, in Estonia, the products were sold as over-the-counter medicines and were advised by a pharmacist if necessary). This statement was replaced with an argument that head lice could be transmitted via shaking hands, which was false, but many people tend to believe it is true.

To translate the statements from English to Estonian, the forward-back translation method was used. A validated translator translated the source language (English) statements into the target language (Estonian). Then, a second translator translated the statements from the target language back into the source language. Then, the authors of the study checked the compatibility of the back translation with the statements of the source language. The questionnaire, in general, was pretested and validated by 16 parents whose children had had head lice, resulting in good levels of understanding, acceptability, and time spent to complete the questionnaire. The average time spent completing the questionnaire was approximately 15 min.

The software programs Sigma Plot for Windows, version 11.0 (GmbH Formation, Germany), and R 2.6.2 (A Language and Environment, http://www.r-project.org, accessed on 10 September 2022) were used. Descriptive statistics were performed; continuous data and proportions were compared using the *t*-test or Mann–Whitney test and chi-square or Fisher’s exact test, respectively. For calculating participants’ knowledge about pediculosis, all correct answers for all participants were summarized (score). To identify potential factors, such as a participant’s age or education (basic, secondary, or high) separately, univariate and multivariate linear regression analyses were performed for healthcare education, living in an urban or rural location, the occurrence of pediculosis in children, the number of children in a family, the children’s institution (preschool or elementary school), which could all affect the transmission of head lice [5] and also might be associated with the score. To identify the potential independent factors associated with the occurrence of pediculosis, the logistic regression analysis was used to adjust for the foregoing factors. A *p*-value of <0.05 was considered to be significant.

## 3. Results

### 3.1. Data of Participants and Occurence of Pediculosis According to Parents

A total of 1503 parents completed the questionnaire, but 15 responses were excluded from the data analysis due to missing data in the responses; therefore, 1488 (1130 and 358, the participants with a child from preschool and elementary school, respectively) questionnaires were included in the study. The main participants were mothers (97.4%). Pediculosis occurred in 34.7% of the children of the study participants, with more occurrence among elementary school children than kindergarten (45.8% vs. 31.2%, *p* < 0.001). The educational and main characteristics are presented in Table 1.

The results of the logistic regression analysis (Table 2) showed that children from elementary school vs. preschool had a greater chance of getting pediculosis. Pediculosis was favored by a higher parental awareness score, a lower level of parental education when compared to secondary education, and higher education. Urban children were less likely to be infected when compared to rural children.

### 3.2. Sources and Management of Pediculosis in Preschool and Elementary School Children

Almost half of the children with pediculosis had experienced it more than twice, and recurring infestations were more common in school-aged children (Table 3). The parents of those children were also better at remembering the source of the contamination. About 40% of infestations occurred in elementary school or preschool, and preschool children got lice more frequently from relatives than from other preschool children. The parents of preschool children informed the school program about having pediculosis more often than the parents of elementary school children. The use of antilice shampoo for treatment was more common among the parents of preschool children.

### 3.3. Parents’ Knowledge of Pediculosis and Influencing Factors

Parents’ awareness of pediculosis was assessed with 20 different statements. Each of the 193 correct answers gave one point, meaning the maximum score was 20 points. The mean score of the 194 correct answers was 14.0 ± 3.4; a total of 1284 (86.3%) participants gave at least more than 50% correct answers. The percentage of correct answers to each question are presented in Table 4. There were six (0.4%) parents who did not give any correct answers.

The results of the univariate linear regression analysis (Table 5) showed that higher age, secondary and higher education (as compared with basic education), healthcare education, and the previous occurrence of pediculosis in a family positively influenced the correct answer scores. The number of children in a family and the area of residence (city vs. rural) did not affect the knowledge status. In the multivariate linear regression analysis, the influencers of better knowledge were the higher age, previous occurrence of pediculosis in the family, and parents’ healthcare education.

## 4. Discussion

Pediculus humanus capitis is transmitted more often in the age group 3–13 years [25,26], especially at the primary school level, i.e., children aged 6–9 years [6]. The children in our study were also of this age, and one-third of them had suffered from pediculosis during their lifetime. The current study also revealed that parents had quite good knowledge about lice infestation, its transmission, and treatment, and this was influenced by the parents’ older age and higher and/or healthcare education level. Raising awareness and schooling about lice is considered to be the most important factor in reducing the prevalence of pediculosis in schools [3]. Unfortunately, our study did not provide the exact sources of where the knowledge was obtained; it can be assumed that the knowledge was obtained from previous experience since half of the children with pediculosis had it more than once.

Given today’s highly developed medical system and presumably better hygienic conditions (than in the past), pediculosis continues to be an important public health concern worldwide [27], although the prevalence may vary from country to country. On the one hand, similar to our study, in Iran, one-third of pupils have been infested with head lice [15]. On the other hand, in Turkey [28] and in Jordan [24], the prevalence of head lice among children has been reported to be twice as low. The prevalence is usually higher in developing countries. For example, Heukelbach & Ugbomoiko [18], in their study in Africa (Nigeria), found that 74% of participants had pediculosis. A similar prevalence was found among children in Brazil [29], Argentina [13], and Thailand [30]. It is also important to point out that most studies did not represent the entire population and, instead, were focused on certain groups of interest [2,5,21], unlike the current study. All the counties in Estonia were involved in our study and the participation rate was high considering the overall population of Estonia (1.3 million) [31].

The results of our study showed that, equally, in preschool and elementary school, about half of the children have had a positive history of pediculosis. This means that head lice are common in this age group; younger children tend to play and socialize in closer physical contact [1]. In addition, this might be due to higher levels of personal hygiene practice among older children. This is supported by a similar result reported by a previous study conducted among school children in Iran [32]. Previously, it has been repeatedly found that the transmission of head lice was more frequent among girls [7,8], probably because hair length plays an important role in the transmission of lice [33], and girls usually have longer hair than boys. However, the results of a study conducted in Norway [22] had already shown that medium-length hair (up to the ears or shoulders) was important for the transmission of pediculosis, and since many boys today also have such long hair, gender differences may no longer play such a role. However, since head lice are highly contagious and are transmitted easily through direct or indirect contact with an infected person and/or infected personal belongings, it is not surprising that pediculosis continues to occur in all age groups and in boys and girls. Unfortunately, this was a shortcoming of our study, as we did not differentiate children by their gender and hair length.

Shampoos that contain pediculicides and hair combing are the most often used treatments for pediculosis. None of our subjects had used oral medication (ivermectin). In Estonia, it is possible to purchase antipediculosis shampoos over the counter, which is a simple (often needed only once) and effective treatment [23]; therefore, the frequent use of this treatment method is completely understandable. However, the relatively high price of shampoos can certainly sometimes become a deterrent when using them, which Gunning et al. [34] and Parison et al. [35] also highlighted in their studies. Special shampoo was also reported as the most used product against head lice in Egypt [36], but over half of the respondents also mentioned picking by hand. In our study, none of the parents used manual picking, which could be due to cultural differences and the better availability of shampoo in Estonia.

Different studies [10,16,17,18,19,37] have shown that the knowledge of pediculosis among parents was fragmentary and should be improved. A study carried out in Brazil on parents’ knowledge of pediculosis found that only one-third of respondents knew correct information about head lice transmission, and even more of the respondents mistakenly correlated control over pediculosis with hygiene. It is also important to point out that there was a significant association between the presence of pediculosis and parents’ poor knowledge regarding transmission. Erroneous knowledge about head lice transmission may continue to exist because of the small size of the insect, which makes it difficult to notice and, in this case, an infestation goes unsuspectedly untreated [29]. When compared with the results of a study conducted with a methodology similar to our study [24], where only one-third of the respondents provided correct answers to half of the statements, we considered our subjects’ awareness of pediculosis to be good. For example, the knowledge of using hot water to wash clothes and bed linens to kill head lice was, in general, present among our subjects. As head lice can be present on inanimate fomites, it is necessary to heat potentially infested clothing and bedding with hot water to kill all stages of the lice [3]. A similar level of parental awareness has also been shown by studies in Australia [16] and Norway [17], which may indicate better awareness or the better availability of information among the citizens of developed countries.

As we previously presented, an association between pediculosis transmission and poor knowledge has been shown. In our study, the children whose parents had better knowledge were more at risk of lice infestation. At this point, we do not think that the risk lies with the parents’ knowledge, but rather that their previous contact with lice put them in a situation where they had to look for information about lice and, thus, they gained awareness. Understandably, parents with medical education and older parents also had more knowledge. The latter may correlate with life experience. Previous studies had shown that children’s infestations were significantly reduced when their mothers were highly educated [38,39]. The prevalence of infestations has also been found to be associated with low educational levels (of parents) in a study conducted in Iran [40]. Thus, based on the results of our study, we can state that parents’ knowledge does not contribute to the prevention of pediculosis. They are definitely better at recognizing and dealing with problems, but their knowledge is probably gained from experience.

Unfortunately, we did not look at children’s awareness and education about pediculosis in schools. However, for example, in a study conducted in Poland [6], it was found that as many as 95.2% of the schools that participated in the study (*n* = 168) discussed issues related to the transmission of lice, although usually only when the problem had already arisen, and not proactively. Healthcare education as early as elementary school can influence health by increasing behaviors that are key to reducing the prevalence of head lice infestation. Therefore, children should be told about pediculosis proactively and not wait until they become infected. In addition to raising the awareness of children and their parents, the effective control of head lice requires the monitoring of children’s contacts and synchronized treatment [16], which is based on rapid information exchanges among school, parents, playmates, and fellow students [23,41].

In summary, we agree with our Polish colleagues [6], who concluded that the lack of legal obligations to report pediculosis outbreaks had a negative impact on the rate and scope of therapeutic activities. Training children and parents on the subject of pediculosis must be consistent, regardless of whether or not there is a current infestation of head lice. In addition to training, however, national screenings are also needed. In order to control the spread of head lice, there is a need to regularly check the presence of head lice in children using appropriate methods, especially during the peak season of their transmission [16,42], which is late summer and autumn in Europe [43].

The study had some limitations. Firstly, we used only one (instead of two) translators to translate the questionnaire. In our study, we did not look at the socioeconomic characteristics of the parents, nor did we specify the gender of the children and how long their hair was. It was certainly possible to investigate the actual occurrence of pediculosis in children, but it would have only provided information about the current situation, and that was not the aim of the study. Despite the shortcomings, in the context of Estonia, there were enough respondents to the questionnaire to conclude that pediculosis is still relevant in our country. Additionally, it can also be considered that not all parents may be able to identify pediculosis, and some parents may have mistaken dandruff for lice. It can be assumed that respondents who had addressed the topic and who had been in contact with pediculosis were more likely to answer the questionnaire. Therefore, we can hope that, in reality, less than one-third of Estonian children have suffered from pediculosis.

## 5. Conclusions

The results of our study show that, despite parental awareness, pediculosis infestations continue to be prevalent among our children, and therefore we will continue to raise the question of how we can more effectively limit the transmission of head lice. It is possible that the restoration of a national screening system would be especially helpful during the most active period of lice transmission. Different campaigns and advertising posters in cityscapes and social media could be pointed out, similar to vaccinations and the dangers of smoking, etc. Close cooperation between the Board of Health and children’s institutions and between children’s institutions and parents is also important. Family nurses and family doctors, to whom parents often turn with their concerns, should also pay more attention to the problem.

## Figures and Tables

**Table 1 medicina-58-01773-t001:** Sociodemographic data and basic pediculosis prophylactic practices for all study participants.

Parameters		All Observed Children				Preschool Children			Elementary School Children		
		All	With Pediculosis	Without	*p*=	With Pediculosis	Without	*p*=	With Pediculosis	Without	*p*=
*n*=		1488	516	972		352	778		164	194	
Parents’ age (mean; SD)		33.9; 6.2	35.5; 6.1	32; 6		34.2; 6	32.1; 5.6		38.1; 5.7	36.8; 6.4	
Education	Basic	8.3	10.8	6.9	0.011	12.5	7.7	NS	7.3	3.6	NS
	Secondary	47.9	45.9	49.0	NS	51.4	51.6	NS	34.1	38.4	NS
	High	43.8	43.2	44.1	NS	36.1	40.6	NS	58.5	58.2	NS
Healthcare education		14.4	13.4	15.0	NS	14.4	14.8	NS	17.7	16.0	NS
Living	Urban	62.8	59.3	64.7	0.046	59.1	63.1	NS	59.8	71.1	0.031
	Rural	36.0	39.7	33.9	0.031	39.8	35.2	0.03	39.6	28.9	0.042
Other children in family in age <10 years	1 child	67.4	68.4	66.9	NS	60.8	61.6	NS	84.8	88.1	NS
	2	28.6	28.1	28.9	NS	34.4	33.3	NS	14.6	11.3	NS
	3	3.6	3.3	3.8	NS	4.5	4.8	NS	0.6	0	NS
	4	0.1	0	0.2	NS	0	0.3	NS	0	0	NS
	≥5	0.2	0.2	0.2	NS	0.3	0.1	NS	0	0.5	NS
Parent checks the child’s hair from time to time	Yes	60.0	72.5	53.4	<0.001	72.7	54.4	<0.001	72.0	49.5	<0.001
	No	37.8	25.8	44.2	<0.001	25.3	43.4	<0.001	26.8	47.2	<0.001

NS—not significant difference (*p* > 0.05).

**Table 2 medicina-58-01773-t002:** Factors influencing the occurrence of pediculosis in children (results of univariate logistic regression analysis).

Factor	OR=	95% CI
Higher score for knowledge	1.26	1.21–1.31
Children in elementary vs. preschool	1.87	1.47–2.38
Parent’s basic vs. secondary education	1.68	1.14–2.47
Parent’s basic vs. higher education	1.61	1.09–2.37
Living in urban vs. rural areas	0.79	0.64–0.99

**Table 3 medicina-58-01773-t003:** Sources and management for lice in families with a history of pediculosis.

		Preschool	Elementary School	
*n*=		352	164	p=
Number of pediculosis in particular children	1	53.4	49.4	NS
	2	22.2	24.4	NS
	≥3	9.4	16.5	0.028
	No data	0.9	0.6	NS
Remembering the source of pediculosis		25.3	70.1	<0.001
Source of pediculosis	Elementary school	0	42.7	<0.001
	Preschool	45.2	11.0	<0.001
	Holiday camp	0.6	0.6	NS
	Training	0.9	2.4	NS
	Friend	11.9	13.4	NS
	Public transport	0.3	0.6	NS
	Relative	10.8	1.8	<0.001
Reporting school about pediculosis	Yes	81.25	68.3	0.002
	No	8.5	22.6	<0.001
	Not remember	1.1	4.9	0.021
Knowledge about previous pediculosis in school	Yes	40.9	42.1	NS
	No	37.2	40.0	NS
	Not remember	6.25	6.7	NS
Treatment of pediculosis		98.6	100	NS
For treatment was used	Head lice shampoo	82.4	74.4	0.047
	Lice comb	84.7	81.1	NS
	Electric lice comb	19.9	25.6	NS
	Head lice oil	30.7	28.0	NS
	Head solution	16.2	12.8	NS
	Other	8.0	3.0	0.054
Effectuality of the method	Yes	94.0	97.0	NS
	No	5.1	3.0	NS
	Not remember	0.3	0	NS
Prophylactic treatment of other members of family	Yes	80.4	73.2	NS
	No	18.2	25.6	NS
	Not remember	0.9	1.2	NS

NS—not significant difference (*p* > 0.05).

**Table 4 medicina-58-01773-t004:** Data awareness of Estonian parents about pediculosis and its prevalence.

Questions	% of Correct Answers
Head lice are parasitic insects	71.4
Head lice live and feed only on human scalps	64.5
Head lice can live up to 1 year on a person’s head	19.8
If a louse falls off a person, it dies within 2–3 days	40.5
Eggs hatch in 10–14 days and can reach reproductive maturity within 8–10 days	65.4
Nits are viable for up to 2 weeks	75.4
Head lice are most often found toward the nape of the neck	37.4
Head lice can fly from person to person	57.7
Head lice eggs slide off easily	83.2
Head lice infestations occur only in developing countries	91.0
Boys get head lice more frequently than girls	69.8
The first symptom of having head lice is scratching the head	84.9
Scratching the head may lead to skin infections	89.7
Head lice can be contracted from animals	53.6
Prolonged direct contact with an infested person is one of the main modes of head lice transmission	63.2
Head lice are transmitted by shaking hands	78.2
Sharing infested combs/brushes does not result in transmission	94.2
During infestation, you should delouse your house by using hot water to wash all clothing and bedding	97.7
Contracting head lice is always a sign of poor hygiene	90.5
You should use extra amounts of lice-killing medication to get the best result	73.9

**Table 5 medicina-58-01773-t005:** Factors affecting the score of the correct answers to statements about pediculosis (univariate and multiple linear regression analysis).

Factor	Coefficient	*p*=
Univariate linear regression analysis		
Previous pediculosis in family	2.09	<0.001
Child in elementary school vs. preschool	0.58	0.005
Parent’s secondary vs. basic education	0.7	0.033
Parent’s high vs. basic education	0.74	0.026
Parent’s healthcare education	1.19	<0.001
Parent’s higher age	0.11	<0.001
Multiple linear regression analysis		
Previous pediculosis in family	0.07	<0.001
Parent’s healthcare education	1.19	<0.001
Parent’s higher age	1.15	<0.002

## Data Availability

Not applicable.

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
