# Peer review of "Estonian Parents’ Awareness of Pediculosis and Its Occurrence in Their Children"

_medicina, 2022, doi:10.3390/medicina58121773_

Round 1

Reviewer 1 Report

It seems that there are points in this article that should be given more attention and scrutiny by respected authors. Below are some of these points:

1. In this study, there was a comparison between primary school and preschool children, but in lines 157 to 159, a comparison was made between kindergarten children and preschool children, which I think was a typing error.

2. As we know, translation from English to Estonian should be called forward translation and translation from Estonian to English should be called backward translation. Unfortunately, in line 111, the author mistakenly stated the translation from English to Estonian as backward translation. In this study, the researcher has used the processes of forward-backward translation, not only backward translation in the method.

3. In addition, according to the instructions, both forward and backward translation processes should be done by at least two people, while in this study it was done by only one person. Therefore, this point should be mentioned as one of the limitations of the study.

4. The explanations related to the studied sample are not enough. For example, it must be stated that the study was conducted in the whole country of Estonia, and sampling method in the study method.

5. How was the pediculosis treatment variable (Table 2) measured in this study? Please clarify.

6. How was “the healthcare education” (Table 1) measured in this study? For example by year, Please clarify.

7. There are some incomprehensible sentences and phrases in the article that need changes in writing in order to be understood; for example “Prophylactic children hear control” in Table 1; or these lines “The mean score of correct answers was 14.0 ± 3.4.17 (1.1%) respondents who achieved 168 a maximum score (20), among which seven of the respondents had healthcare education; 1284 (86.3%) participants gave, at least, more than 50% correct answers.”

8. The same problems in understanding can be seen in many parts of the article, including the title “the knowledge of pediculosis”.

9. What’s “3.4.17” in line 168?

Good luck to the authors of the article

Author Response

Comments and Suggestions for Authors

It seems that there are points in this article that should be given more attention and scrutiny by respected authors. Below are some of these points:

  1. In this study, there was a comparison between primary school and preschool children, but in lines 157 to 159, a comparison was made between kindergarten children and preschool children, which I think was a typing error.

Thank you for this excellent remark. The corrections are done. Kindergarten was replaced by preschool throughout the text.

  1. As we know, translation from English to Estonian should be called forward translation and translation from Estonian to English should be called backward translation. Unfortunately, in line 111, the author mistakenly stated the translation from English to Estonian as backward translation. In this study, the researcher has used the processes of forward-backward translation, not only backward translation in the method.

Thank you, it was a mistake on our part. A forward-backword translation method has been used, and the limitation of our work is the use of only one translator. Corresponding additions made in the manuscript.

  1. In addition, according to the instructions, both forward and backward translation processes should be done by at least two people, while in this study it was done by only one person. Therefore, this point should be mentioned as one of the limitations of the study.

See the answer to the previous question.

  1. The explanations related to the studied sample are not enough. For example, it must be stated that the study was conducted in the whole country of Estonia, and sampling method in the study method.

Thank you, text corrected accordingly: “The study was conducted in two stages in whole country of Estonia (non-probability sampling).”

  1. How was the pediculosis treatment variable (Table 2) measured in this study? Please clarify.

It was an open question and the respondent could suggest several methods. Similar responses were summed. 

  1. How was “the healthcare education” (Table 1) measured in this study? For example by year, Please clarify.

 Thank you for this question; it was a so-called closed question "Do you have a healthcare education?" (no/yes answer).

  1. There are some incomprehensible sentences and phrases in the article that need changes in writing in order to be understood; for example “Prophylactic children hear control” in Table 1; or these lines “The mean score of correct answers was 14.0 ± 3.4.17 (1.1%) respondents who achieved 168 a maximum score (20), among which seven of the respondents had healthcare education; 1284 (86.3%) participants gave, at least, more than 50% correct answers.”

 Thank you; it is really a confusion on our part. Text corrected accordingly:

- The parent checks the child's hair from time to time.

- Parents' awareness of pediculosis was assessed with 20 different statements. Each correct answer gave one point, so the maximum score was 20 points. The mean score of correct answers was 14.0 ±3.4; 1284 (86.3%) participants gave, at least, more than 50% correct answers.

  1. The same problems in understanding can be seen in many parts of the article, including the title “the knowledge of pediculosis”.

 Täname, pealkiri täpsustatud: “Parents` Knowledge of Pediculosis and Influencing Factors”

  1. What’s “3.4.17” in line 168?

Thank you, it's really a confusion on our part. Text corrected accordingly: “The mean score of correct answers was 14.0 ±3.4…”

Good luck to the authors of the article

Thank you for your excellent comments!

Reviewer 2 Report

The text must be completely revised and transformed into a version adapted for scientific publication. The tables are not informative, with errors and difficult to understand.

In materials and methods it is reported that all Estonian parents were the target of the study and after that the questionnaire was sent to the city of Tartu and its surroundings. Only cite the bibliography that was used for the quationary and not report on the author and how the English translation was done.

 Be careful when stating the prevalence that should be placed as reality and not the one that was pointed out by the parents. In table 1 there are conceptual errors, note that it appears in the first line as the average age of the participants, which appears to be the prevalence obtained. Descriptions of all tables should be self-explanatory and very clear.

Discussion is 23 lines before starting discussion of manuscript results. In addition, in the discussion the wording follows very elementary as in the introduction and methods.

Therefore, I suggest that the text be completely revised and resubmitted.

Author Response

The text must be completely revised and transformed into a version adapted for scientific publication. The tables are not informative, with errors and difficult to understand.

Thank you for your thought provoking comment. We compared our text and tables with other similar articles (ALBashtawy et al. 2012, Counahan et al. 2007, DeSouza et al. 2022) and it remains unclear what exactly was meant. Pediculosis can be considered a so-called soft branch in medicine, and thus the wording is less medical. The manuscript has also been linguistically checked by MDPI translators. Similar to the text and wording, we also consider the tables in our manuscript to be analogous to other articles. We adjusted the Table 1 and hope that by "error" you meant the changed part of the table. In addition, we added two tables to better highlight the results.

In materials and methods, it is reported that all Estonian parents were the target of the study and after that the questionnaire was sent to the city of Tartu and its surroundings. Only cite the bibliography that was used for the quationary and not report on the author and how the English translation was done.

Thank you for your comment. The questionnaire was distributed in various family school forums, which made it possible to involve parents from all over Estonia. In addition, it was used in the lists of Tartu and its immediate surroundings. The results show that there were respondents from all over Estonia, primarily from Tallinn and Tartu, but these two regions also make up 60% of the entire country's population. However, since the regions are very country-specific, we considered it necessary to delete this note in terms of methodology altogether.

When using questionnaires, it is always important to highlight its preparation or, in this case, its translation and adaptation. Therefore, we consider it necessary to describe our methodology as precisely as possible. We agree to delete lines 126-133, but would leave that final decision to the Editor. 

Be careful when stating the prevalence that should be placed as reality and not the one that was pointed out by the parents.

Thank you for the excellent note! Text modified accordingly:

  • Title: Estonian Parents' Awareness of Pediculosis and Its Occurrence in Their Children
  • Abstract: The purpose of this study was to clarify the occurence of pediculosis among Estonian preschool and primary school-aged children according to their parents and their parents' awareness of pediculosis and related behaviors. According to parents, pediculosis had occurred in 34.7% of children, and more than one-third of pediculosis patients had experienced it more than twice.
  • Methods: To identify the potential independent factors associated with the occurence of pediculosis, the logistic regression analysis adjusted for the foregoing factors was conducted.
  • Results: 2. Data of Participants and Occurence of Pediculosis According to Parents

In table 1 there are conceptual errors, note that it appears in the first line as the average age of the participants, which appears to be the prevalence obtained. Descriptions of all tables should be self-explanatory and very clear.

Thank you for your comment! We understand your point and hopefully we have now expressed ourselves better in Table 1.

Discussion is 23 lines before starting discussion of manuscript results. In addition, in the discussion the wording follows very elementary as in the introduction and methods.

Thank you for your comment, but unfortunately we cannot accept this comment totally. The discussion begins with highlighting the most important results (In first paragraph: …parents had quite good knowledge; …were also of this age, and one-third of them had suffered from pediculosis during their lifetime…) comparing with the other studies; so in our opinion, the discussion in the first paragraph is also presented in the light of the results of our own study.

In the second paragraph we tried to number Estonia among the other countries by the occurrence of pediculosis (On the one hand, similar to our study, in Iran, one-third of pupils have been infested with head lice...)

Thank you for your excellent remarks!

Reviewer 3 Report

This manuscript represents a research article that discussed Estonian parents' awareness of pediculosis and its prevalence. There are some concerns in this manuscript as follows:

1.    Key words: The word “Estonian” should be included in the key words.

2.    The novel points in this article should be clarified in the “Introduction” section.

3.    Although this research article is about Estonian parents' awareness of pediculosis and Its prevalence, the “Introduction” in page 2 Line 60 talks about a Nigerian study. The awareness and prevalence differ from one population to another. Please, be focused on the studies that are related to the Estonian and their similar populations.

4.    Page 2 the last paragraph: It should be in the Past tense; i.e. “examines” should be replaced with ”examined” and ”is aimed” should be replaced with ”aimed”.

5.    Regarding the questionnaire used, test–retest reliability should be calculated using Spearman’s correlation coefficient (r).

6.    A collective diagram is recommended to summarize the main findings of the current study.

7.    I think that the conclusion was not sufficient. All possible solutions to the problem of pediculosis should be mentioned. Also, the clinical implications of the data obtained from the present study should be mentioned.

8.    The limitations of the study should be mentioned.

9.    The manuscript should be thoroughly checked regarding the grammatical and typing errors.

Author Response

Comments and Suggestions for Authors

This manuscript represents a research article that discussed Estonian parents' awareness of pediculosis and its prevalence. There are some concerns in this manuscript as follows:

  1. Key words: The word “Estonian” should be included in the key words.

Thank you for the excellent note, correction done!

  1. The novel points in this article should be clarified in the “Introduction” section.

Unfortunately, we don't understand what you mean exactly. In the Introduction chapter, the topic is opened in the light of the results of other similar studies. The purpose of our study and the rationale for why we consider it necessary to study the topic are also attached.

However, we added a justification at the end of the Introduction chapter regarding the necessity of researching the topic (novelty):

„… Corresponding results are extremely necessary at the national level for the Board of Health, which can plan further activities according to parents' awareness or lack of knowledge (for example, restoring the registration of pediculosis cases), as well as for family doctors and family nurses, whose task is primary counselling of parents (including about pediculosis).

  1. Although this research article is about Estonian parents' awareness of pediculosis and Its prevalence, the “Introduction” in page 2 Line 60 talks about a Nigerian study. The awareness and prevalence differ from one population to another. Please, be focused on the studies that are related to the Estonian and their similar populations.

Thank you for the excellent note! We totally agree with that, and so we've added what we think is an important point at the end of the paragraph:

In Nigeria, 496 people were examined for the presence of head lice and their awareness of its transmission. The results showed that 74% of the respondents had suffered from pediculosis, but only 11.1% of them knew how head lice was transmitted. Combing was most often mentioned as a treatment method (46.3%), and only 4.6% of the respondents had used lice medications  [19]. Another survey about parents’ awareness showed that certain beliefs and perceptions regarding head lice generated worry and confusion in parents as well as unhealthy actions to combat infestations. It was also revealed that there was an inverse correlation between parents’ educational level and the number of pediculosis cases [20]. It has also been found that maternal education, knowledge, and attitude towards pediculosis capitis infestation are significantly associated with the prevalence of pediculosis [8]. The same factors and also the number of times combing hair per day were pointed out as risk factors for pediculosis in another study [21]. One study revealed a greater association between the prevalence of infestations and individuals who already had a previous history of pediculosis and who had suffered from head itching. There was a significant relationship among pediculosis, head itching, and a previous history of pediculosis [22]. The previously mentioned studies were carried out in Africa, South America or the Middle East, and thus one may doubt whether these results are suitable for the Estonian cultural space. For example, the results of studies conducted in Northern Europe (Norway) show that lice may be less common among children of the same age, but still abundant (about 45% of children) [], and in addition to combing, pediculicides are also used to treat pediculosis, but parents' awareness of pediculosis is still important for identifying and dealing with lice []. So it seems that regardless of geographic location, what is associated with lice is still similar as they are highly contagious regardless of country.

  1. Page 2 the last paragraph: It should be in the Past tense; i.e. “examines” should be replaced with ”examined” and ”is aimed” should be replaced with ”aimed”.

Thank you for the excellent note, corrections done!

  1. Regarding the questionnaire used, test–retest reliability should be calculated using Spearman’s correlation coefficient (r).

As our study was carried out in two stages, but on different sample (test-retest variability is possible when the methods, sample and other conditions are identical), the test-retest method here is not possible to use to prove the reliability. The two stages of the study were not same and our aim was not to get same results as test-retest assumes, but to find possible differences in two different samples (parents of preschool vs elementary school children). It is also possible, that we did not understand your suggestion very clearly.

  1. A collective diagram is recommended to summarize the main findings of the current study.

Thank you for the excellent note! Thus, we added two tables to the manuscript: Table 2 reflects the factors influencing the occurrence of pediculosis, and Table 5 reflects the factors influencing the correct answer score. Also, we added in table 5 – previous pediculosis in the family – as the influencing factor of the parents` correct answer. The text has also been adjusted accordingly.

  1. I think that the conclusion was not sufficient. All possible solutions to the problem of pediculosis should be mentioned. Also, the clinical implications of the data obtained from the present study should be mentioned.

Thank you for the excellent note! Conclusion has been supplemented as follows:

The results of our study show that despite parental awareness, pediculosis infesta-tions continue to be prevalent among our children, and therefore, we continue to raise the question of how we can more effectively limit the transmission of head lice. It is possible that the restoration of a national screening system would be especially helpful during the most active period of lice transmission. Different campaigns and advertising posters in cityscapes and social media could be pointed out, similar to vaccinations, the dangers of smoking, etc. Close cooperation between the Board of Health and children's institutions and between children's institutions and parents is also important. Family nurses and family doctors, to whom parents often turn with their concerns, should also pay more at-tention to the problem.

  1. The limitations of the study should be mentioned.

Thank you for excellent remark. The text has been supplemented as follows:

The study had some limitations. Firstly, we did not look at the socioeconomic char-acteristics of the parents, we did not specify the gender of children, and how long their hair was. It was certainly possible to investigate the actual occurrence of pediculosis in children, but it would have provided information only about the current situation and that was not the aim of the study. Despite the shortcomings, in the context of Estonia, there were enough respondents to the questionnaire to conclude that pediculosis is still relevant in our country. Although it can also be thought that not all parents may be able to identify pediculosis, and some parents may have mistaken dandruff instead for lice. It can be as-sumed that respondents who had addressed the topic and who had been in contact with pediculosis were more likely to answer the questionnaire. Therefore, we can hope that, in reality, less than one-third of Estonian children have suffered from pediculosis..

  1. The manuscript should be thoroughly checked regarding the grammatical and typing errors.

Thank you for your prompt feedback. The manuscript has already been pre-reviewed by MDPI's language proofreader and so we are confused as to whether we need to order another language check from another provider.

Thank you for your excellent remarks!

Round 2

Reviewer 1 Report

Unfortunately, all problems are still present in the corrected attachment.

Author Response

Dear experts, we are very sorry for the confusion. In the PDF version, the changes were clearly visible, but in the WORD version, all changes were not saved. We hope that the version presented is now correct. We apologize for the extra time this has caused you.

Reviewer 2 Report

Version 2 attached for review seems to me to be identical to version 1. 

I didn't receive the version with the modifications indicated by the authors, so I cannot adequately review the publication.

I pointed out in the attached file some notes.

Author Response

(The authors gave the same response as above.)

Round 3

Reviewer 1 Report

Dear Editor,

Thank you for the opportunity to review the article again. Unfortunately, the answers of the respected authors of the article to comments 5 and 6 in the previous review are not clear and there is a need for more explanations.

Regards

Author Response

Thank you for the opportunity to review the article again. Unfortunately, the answers of the respected authors of the article to comments 5 and 6 in the previous review are not clear and there is a need for more explanations.

Regards

  1. How was the pediculosis treatment variable (Table 2) measured in this study? Please clarify.

This information is presented in Table 3 and the question in our questionnaire was presented as follows:

"What kind of means/measures did you last use to treat pediculosis (you can choose more than one if you used different methods)?"

Options: skin solution, lice shampoo, lice oil, lice comb, electric lice comb, other (specify)! Similar responses were summed. Table 3 reflects data only for those whose children have had pediculosis. The percentage is therefore calculated from all those who had pediculosis. It turns out that several different methods are used simultaneously to treat lice.

We also changed the title of the table and hopefully now its content is also more understandable: “Sources and management with lices in families with a history of pediculosis.”

  1. How was “the healthcare education” (Table 1) measured in this study? For example by year, Please clarify.

The question was posed as follows: "Do you have a higher education in the field of healthcare?" We were not interested in the exact specialty or length of service, but only information on whether or not there is a corresponding education. In our context, it could be, for example, a nurse, midwife, physiotherapist, doctor, etc. We would like to clarify that the word "higher" was previously omitted in our answer because it was lost in translation.

Thank you for these excellent comments!

Reviewer 2 Report

The manuscript is of high importance and has had major changes and after reading the justifications I agree with the publication after minor revisions.

 Below I transcribe the responses of the authors and highlight my revisions.

Thanks for your thought provoking comment. We have compared our text and tables with other similar articles (ALBashtawy et al. 2012, Counahan et al. 2007, DeSouza et al. 2022) and it is still not clear what exactly is meant. Pediculosis can be considered a so-called mild branch of medicine, and therefore the wording is less medical.

  Paragraphs are very long and written in the form of a monograph or dissertation. The editors of the Journal of Medicina must confirm the form and accept it in the form of this manuscript. If accepted by the editors I withdraw my comments.

The manuscript was also linguistically checked by MDPI translators.

If it is approved by the Medicina journal, I accept it and perhaps my criticism came from the paragraphs and very long sentences, which is not common in the English language.

Similar to text and wording, we also consider the tables in our manuscript to be analogous to other articles. We tweaked Table 1 and hopefully by "error" you mean the changed part of the table. In addition, we added two tables to better show the results.

Table 1 is clearer, but both this and the others, the title must be self-explanatory and must be completed with data from the title of the manuscript:

Ex.Data Awareness of Estonian parents about pediculosis and its prevalence.

This is the only point that really needs to be revised by the authors of the manuscript. The other points the editors must approve or not.

In materials and methods, it is reported that all Estonian parents were the target of the study and after that the questionnaire was sent to the city of Tartu and its surroundings. Cite only the bibliography that was used for the questionnaire and do not inform about the author and how the translation into English was made.

Thanks for your comment. The questionnaire was distributed in various school family forums, which enabled parents from all over Estonia to be involved. Furthermore, it was used in the lists of Tartu and its immediate surroundings. The results show that there were respondents from all over Estonia, mainly from Tallinn and Tartu, but these two regions also account for 60% of the entire country's population. However, as regions are very country-specific, we consider it necessary to completely exclude this note in terms of methodology.

Now I understand and the deletion of some data made it easier for those who don't know Estonia to understand.

When using questionnaires, it is always important to highlight their elaboration or, in this case, their translation and adaptation. Therefore, we consider it necessary to describe our methodology as accurately as possible. We agreed to delete lines 126-133, but would leave that final decision to the Editor.

Ok, but I still think that the whole description is not necessary, but only the reference citing that it was made with modifications. I agree that the decision is up to the editors.

Be careful when stating the prevalence that should be placed as reality and not the one that was pointed out by the parents.

The prevalence of pediculosis, as it is now described, is clearly that described by the parents and that it was not observed by the researchers.

Author Response

The manuscript is of high importance and has had major changes and after reading the justifications I agree with the publication after minor revisions.

 Below I transcribe the responses of the authors and highlight my revisions.

Thanks for your thought provoking comment. We have compared our text and tables with other similar articles (ALBashtawy et al. 2012, Counahan et al. 2007, DeSouza et al. 2022) and it is still not clear what exactly is meant. Pediculosis can be considered a so-called mild branch of medicine, and therefore the wording is less medical.

  Paragraphs are very long and written in the form of a monograph or dissertation. The editors of the Journal of Medicina must confirm the form and accept it in the form of this manuscript. If accepted by the editors I withdraw my comments. 

The manuscript was also linguistically checked by MDPI translators. 

If it is approved by the Medicina journal, I accept it and perhaps my criticism came from the paragraphs and very long sentences, which is not common in the English language. 

Similar to text and wording, we also consider the tables in our manuscript to be analogous to other articles. We tweaked Table 1 and hopefully by "error" you mean the changed part of the table. In addition, we added two tables to better show the results. 

Table 1 is clearer, but both this and the others, the title must be self-explanatory and must be completed with data from the title of the manuscript:

Ex.Data Awareness of Estonian parents about pediculosis and its prevalence.

This is the only point that really needs to be revised by the authors of the manuscript. The other points the editors must approve or not.

Thank you for your comment and great suggestion! Table headings have been revised and corrected as follows:

Table 1: Sociodemographic data and basic pediculosis prophylactic practices of all study participants.

Table 2: -

Table 3: Sources and management with lices in families with a history of pediculosis.

Table 4: Data Awareness of Estonian parents about pediculosis and its prevalence.

Table 5: Factors affecting the score of correct answers to statements about pediculosis (univariate and multiple linear regression analysis)

In materials and methods, it is reported that all Estonian parents were the target of the study and after that the questionnaire was sent to the city of Tartu and its surroundings. Cite only the bibliography that was used for the questionnaire and do not inform about the author and how the translation into English was made.

Thanks for your comment. The questionnaire was distributed in various school family forums, which enabled parents from all over Estonia to be involved. Furthermore, it was used in the lists of Tartu and its immediate surroundings. The results show that there were respondents from all over Estonia, mainly from Tallinn and Tartu, but these two regions also account for 60% of the entire country's population. However, as regions are very country-specific, we consider it necessary to completely exclude this note in terms of methodology.

 Now I understand and the deletion of some data made it easier for those who don't know Estonia to understand.

When using questionnaires, it is always important to highlight their elaboration or, in this case, their translation and adaptation. Therefore, we consider it necessary to describe our methodology as accurately as possible. We agreed to delete lines 126-133, but would leave that final decision to the Editor.

Ok, but I still think that the whole description is not necessary, but only the reference citing that it was made with modifications. I agree that the decision is up to the editors.

Thank you for your comment! If the Editor agrees with the recommendation, we will significantly shorten the text and use the necessary references. 

Be careful when stating the prevalence that should be placed as reality and not the one that was pointed out by the parents.

The prevalence of pediculosis, as it is now described, is clearly that described by the parents and that it was not observed by the researchers.

Thank you for the excellent comments!
